# Analysis of rDNA reveals a high genetic diversity of *Halophila major* in the Wallacea region

Xuan-Vy Nguyen[1,2]☯*, Nhu-Thuy Nguyen-Nhat[1], Xuan-Thuy Nguyen[1‡], Viet-Ha Dao[1,2], Lawrence M. Liao[3‡], Jutta Papenbrock[4☯]

**1** Department of Marine Botany, Institute of Oceanography, Vietnam Academy of Science and Technology, Nha Trang, Viet Nam, **2** Faculty of Marine Science and Technology, Graduate University of Science and Technology, Cau Giay, Ha Noi, Viet Nam, **3** Graduate School of Integrated Sciences for Life, Hiroshima University, Hiroshima, Japan, **4** Institute of Botany, Leibniz University Hannover, Hannover, Germany

☯ These authors contributed equally to this work.
‡ These authors also contributed equally to this work.
* nguyenxuanvi@gmail.com

**Data Availability Statement:** All relevant data are within the paper and its Supporting Information files.

## Abstract

The genus *Halophila* shows the highest species diversity within the seagrass genera. Southeast Asian countries where several boundary lines exist were considered as the origin of seagrasses. We hypothesize that the boundary lines, such as Wallace's and Lydekker's Lines, may act as marine geographic barriers to the population structure of *Halophila major*. Seagrass samples were collected at three islands in Vietnamese waters and analyzed by the molecular maker ITS. These sequences were compared with published ITS sequences from seagrasses collected in the whole region of interest. In this study, we reveal the haplotype and nucleotide diversity, linking population genetics, phylogeography, phylogenetics and estimation of relative divergence times of *H. major* and other members of the *Halophila* genus. The morphological characters show variation. The results of the ITS marker analysis reveal smaller groups of *H. major* from Myanmar, Shoalwater Bay (Australia) and Okinawa (Japan) with high supporting values. The remaining groups including Sri Lanka, Viet Nam, the Philippines, Thailand, Malaysia, Indonesia, Two Peoples Bay (Australia) and Tokushima (Japan) showed low supporting values. The Wallacea region shows the highest haplotype and also nucleotide diversity. Non-significant differences were found among regions, but significant differences were presented among populations. The relative divergence times between some members of section *Halophila* were estimated 2.15–6.64 Mya.

## Introduction

Seagrasses are a polyphyletic group of monocotyledonous angiosperms that play an important ecological role and provide important ecosystem services in various coastal regions [1]. Approximately 72 seagrass species have been identified around the world [2]. Among six global regions, the Indo-Pacific region shows the largest number of seagrass species worldwide

**Funding:** This study was funded by the National Foundation for Science & Technology Development (NAFOSTED), Viet Nam to XVN with grant number: 106.02-2018.313. The funder had no role in study design, data collection and analysis, decision to publish, or preparation of the manuscript.

**Competing interests:** The authors have declared that no competing interests exist.

with 24 taxa that form vast meadows of mixed species stands [3]. Within the genus *Halophila*, eight sections based on their geographic distribution consisting of approximately 24 species have been described [4]. *Halophila ovalis* (Brown.) Hooker 1858 shows a global distribution whereas other members occur only in specific areas [5]. The *Halophila* section including *H. major* (Zollinger) Miquel 1856, *H. ovalis*, *H. bullosa* (Setch.) Kuo, n. comb., *H. minor* (Zollinger) Hartog 1957, *H. gaudichardii* Kuo 2006, *H. ramamurthiana* (Ravikumar *et* Ganesan) Kuo, n. com, *H. mikii* Kuo 2006, *H. linearis* den Hartog 1957, *H. nipponica* Kuo 2006, *H. okinawensis* Kuo 2006, *H. johnsonii* Eiseman 1980, and *H. madagascariensis* Steud. ex Doty *et* Stone 1967 is known to present one the most complex challenges in plant taxonomy [5].

*Halophila major* differs from closely related species by two main characteristics, the number of cross veins and the ratio of the distance between the intramarginal vein, with the lamina margin at the half-way point along the leaf length [6]. The species commonly occurs in Sri Lanka [7] Japan [8], Australia [8, 9], Southeast Asian countries such as Indonesia [10], Philippines [11], Malaysia, Myanmar [12] and Thailand [8]. In Viet Nam, *H. major* was misidentified as *H. ovalis* in the off-shore islands from Nha Trang Bay [13]. Defining taxonomic boundaries within the *Halophila* section has continued to present a real challenge due to leaf morphological traits that overlap among species and due to a high plasticity within species and even within populations [3]. Therefore, molecular markers could provide promising approaches for an unambiguous classification. Among the markers applied, the nuclear ribosomal internal transcribed spacer (ITS1-5.8S-ITS2) region was used to identify *H. ovalis* and closely related species, and species resolution was higher than by the analysis of the concatenated sequences of genes encoding the large subunit of ribulose-1,5-bisphosphate-carboxylase-oxygenase (*rbc*L) and chloroplast maturase K (*mat*K) [14]. Kurniawan et al. [10] found that *H. major* populations in Indonesia seem to split into two groups based on the ITS marker. In addition, the study of Tuntiprapas et al. [15] revealed four different haplotypes of *H. major* within the Andaman Sea based on the ITS marker. Hence, the diversity of haplotypes may be higher than what we currently know.

For the time-calibrated phylogeny of seagrass, both nuclear (ITS) and chloroplast loci (*rbc*L, *mat*K) were used to estimate the divergence of seagrass taxa. Coyer et al. [16] combined both nuclear and chloroplast loci to show the divergence within the seagrass family Zosteraceae. The result based on multi-locus marker analysis revealed that the most recent common ancestor of the Hydrocharitaceae family existed in Asia during the Late Cretaceous and Palaeocene (54.7–72.6 Mya) [17]. The authors also showed that the divergence time for the *Halophila* genus was 19.41 Mya ago. Recently, Kim et al. [18] indicated that the species *H. nipponica* diverged 2.95 ± 1.08 Mya from *H. ovalis*, and the divergence times for *H. ovalis* and *H. major* were similar, around 3.5 Mya. Our previous study on *H. ovalis* populations along the Egyptian coastline showed that the Red Sea *H. ovalis* populations did not group with the *H. ovalis* assemblage worldwide. *H. major*, *H. ovalis* and *H. ovalis* collected from the Red Sea were sister clades [19].

Among the six defined seagrass bioregions of the world, the Indo-West Pacific bioregion (Bioregion 5) is the largest and most diverse [11]. Within this bioregion, the seagrass beds in Southeast Asia have been separated into 22 marine provinces and ecoregions [20]. The Wallacea is located between the Sunda and the Sahul Shelf. It is a distinct region because it comprises many endemic, drought-tolerant floristic elements. The flora of the two shelves is more homogeneous than the Wallacean flora [21]. Several biogeographic barriers and boundary lines are found in this bioregion. The Sunda Shelf is a barrier that restricts the exchange of fish populations between the tropical Indian Ocean and the Pacific Ocean [22] while Wallace's Lines and the modification of Wallace's Line (Huxley's Line) were considered as boundary line for several marine organisms such as the seagrass species *Syringodium isoetifolium* (Ascherson)

Dandy 1939 [23], terrestrial vertebrates [24], and seagrass-associated fungal communities [25]. However, another seagrass species *Thalassia hemprichii* (Ehrenberg) Ascherson 1871 shows a genetically distinguishable cluster located within the Wallacea [23]. There are no any reports about the genetic structure of seagrass found between Wallacea and the Sahul Shelf. The Lydekker's Line seems to be a marine barrier for the populations of blue swimming crab (*Portunus pelagicus* Linnaeus 1758) between the Sunda Shelf/Wallacea and Sahul Shelf [26]. Oceanic currents can act to both promote and limit gene-exchange. For example, the Kuroshio Current influences genetically homogeneous populations of *Enhalus acoroides* (Linnaeus) Royle 1839 between Yaeyama (Japan) and north-east Philippines [27].

These findings lead to the hypothesis that the seagrass species *H. major* may form a monophyletic group in the interesting area. In this study, we analysed genetic diversity and link population genetics of *H. major* populations in the different bioregions. In addition, divergence times of members of the genus *Halophila* were also estimated.

## Materials and methods

### Sampling and species identification

The seagrass materials were collected at different locations in Viet Nam including Ly Son (15.376˚N; 109.135˚E), Con Dao (08.684˚N; 106.626˚E), and Phu Quoc (10.227˚N; 104.684˚E) Island (Fig 1). Ly Son Islands locate in Central Viet Nam, about 30 km from the shore. The Islands consist of two off-shore volcanic islands in the South China Sea, and a few islets. The previous report of Quang et al. [28] indicated that the distribution of seagrass from this area was 188.9 ha. Con Dao Islands is a national park located in the South of Viet Nam. It consists of 120 km$^2$ of sea area and 14 islands. The seagrass beds from this off-shore islands were estimated at 200 ha, that mainly contribute to the main island [29]. Phu Quoc Islands which are located in the Bay of Thailand in the South of Viet Nam are the biggest island of Viet Nam. Seagrass beds at Phu Quoc are known for the largest area (more than 10,000 ha) and their species diversity compared to the other off-shore islands. In this location, nine species including putative *Halophila ovalis* were recorded [29]. The field surveys from the three above described locations were permitted by the People's Committee of Ly Son, Con Dao and Phu Quoc in response to letters from the Institute of Oceanography, Viet Nam.

In the present study, SCUBA diving and snorkelling were used to collect seagrass samples in deep (6–10 m) and shallow water (1–3 m), respectively. At each site, five different plants were collected. At each site, these five different plants were randomly collected across the beds with a distance of 20–25 m interval between two plants. For each plant, one to two young leaves were fixed in DESS solution (20% dimethyl sulfoxide, 0.25 M disodium EDTA, and saturated NaCl) for DNA extraction and morphological observation. The remaining part was pressed as herbarium voucher specimens. Sample information is presented in S1 Table. Voucher specimens were deposited in the Institute of Oceanography (ION), Nha Trang City, Viet Nam. Specimens were identified using the keys of den Hartog [31], Kuo [4], Kuo et al. [6] and Kurniawan et al. [10]. The morphological characters used for measurements were cross veins (CV), branching cross veins (BCV), space between cross veins (SC), the angle between cross veins and midveins (AG), leaf width (LW), and leaf length (LL).

### DNA extraction, PCR amplification and sequencing

The fixed materials (five plants/site) in DESS solution were separately homogenized in liquid nitrogen by mortar and pestle. Of the finely powdered plant material 100 mg was used for DNA extraction using the Quick-DNA$^{TM}$ Miniprep Plus Kit (Zymo Research, CA, USA) following the manufacturer's instructions. The region selected for PCR amplification was the

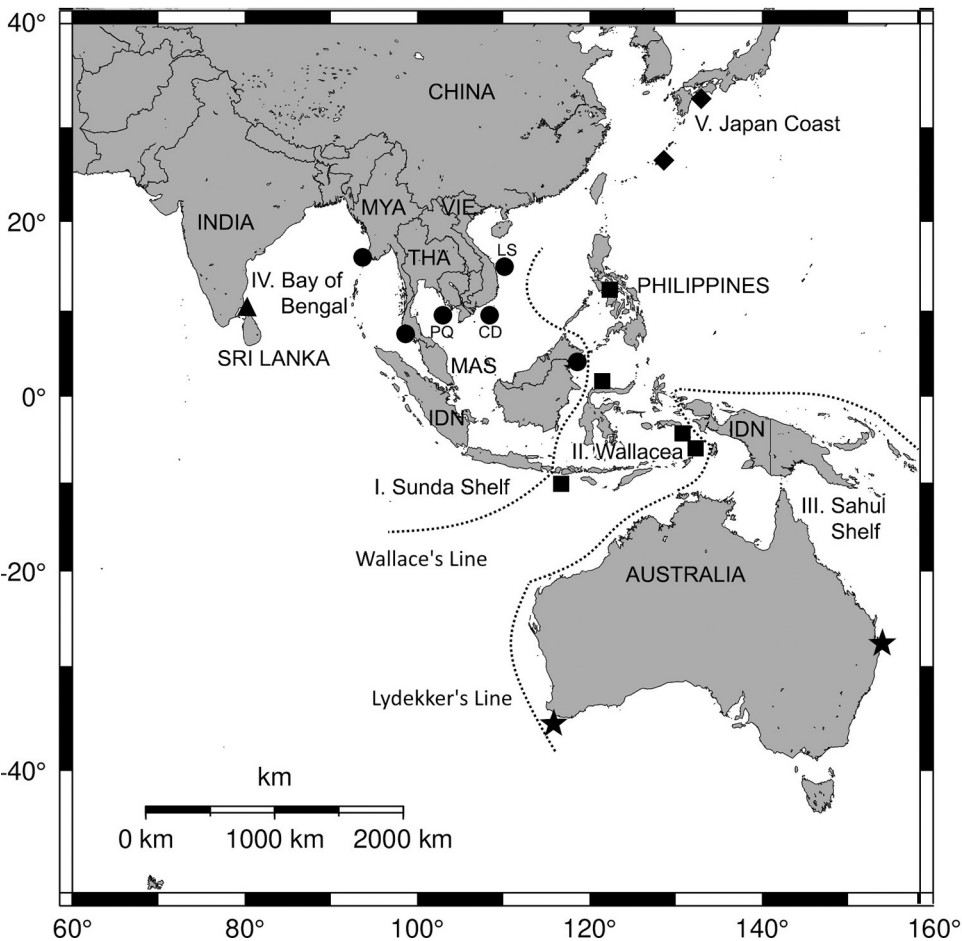

**Fig 1. The map shows the sampling sites in Viet Nam and sequences of other regions were obtained from GenBank.** Region I (Sunda Shelf, sites = solid rounds) includes sampling sites in Viet Nam, Thailand, Malaysia and Myanmar. Region II (Wallacea, sites = solid squares) includes sampling sites in the Philippines and Indonesia. Region III (Sahul Shelf, sites = solid stars) includes sampling sites in Australia. Region IV (site = solid triangle) and V (sites = solid diamond) are sampling sites in the Bay of Bengal and the Coast of Japan, respectively. IDN = Indonesia, MAS = Malaysia, MYA = Myanmar, THA = Thailand, VIE = Viet Nam. LS = Ly Son Island, CD = Con Dao Island, PQ = Phu Quoc Island. Dotted line is the boundary line. Source of digital map: The National Oceanic and Atmospheric Administration (NOAA), USA, public domain data. Wallace's and Lydekker's Lines were adapted from Van Welzen et al. [30].

ITS1-5.8S-ITS2 region. The primers ITS5a (5′ -CCTTATCATTTAGAGGAAGGAG-3′ [32] and ITS4 (5′ -TCCTCCGCTTATTGATATGC-3′ ) [33] were used to amplify sequences of 700 bp. For the ITS1-5.8S-ITS2 region, the total volume of 25 μl included 2x OneTag® Master Mix (New England Biolabs, Ipswich, MA, USA), 10–30 ng template DNA, and 1 pmol of each primer. PCRs consisted of an initial denaturation step at 95˚ C for 4 min, 35 cycles consisting of denaturation at 95˚ C for 40 s, annealing at 52⁰ C for 30 s, and elongation at 72˚ C for 35 s. The 35 cycles were followed by a final extension at 72˚ C for 5 min, terminated by a final hold at 10˚ C. PCR was performed in an Applied Biosystems 2720 thermocycler (Applied Biosystems, Foster, CA, USA) with a heated lid. PCR products were cleaned using a GenElute™ PCR Clean-Up kit (SigmaAldrich, St. Louis, MO, USA) following the manufacturer's instruction. Direct Sanger sequencing of PCR products in both directions was done by 1ST BASE (Selangor, Malaysia) from both directions. The consensus sequence was achieved by Clone Manager 9 (Sci-Ed, Cary, NC, USA).

## Phylogenetic analyses

There were no nucleotide differences in sequence of the five plant samples collected at each site. Therefore, only one of the sequences per site was used in the phylogenetic analysis. For the phylogenetic analysis, the dataset of ITS sequences, including three sequences obtained in this study and 73 sequences of known *Halophila* species retrieved from GenBank (https://www.ncbi.nlm.nih.gov/), were used for analysis (S1 Table). Among them, 69 ITS sequences from *H. major* were produced from samples collected in different regions (Fig 1). The sequences were aligned by the MAFFT algorithm with the selection of the q-ins-i option, considering the secondary structure for the alignment [34]. jModelTest version 2.1.6 [35] and the corrected AIC (Akaike Information Criterion) were used to find the best model for the analysis. *Halophila beccarii* Ascherson 1871 was used as the out-group. Two algorithms including Maximum Likelihood (ML) and Bayesian Inference (BI) were used for the phylogenetic analysis. Phylogenetic analyses were performed using RAxML version 8.1 [36] for Maximum Likelihood (ML) with model parameters fixed according to the values determined. The bootstrap values of the ML tree were estimated via the bootstrap algorithm with 1,000 replications. BI analyses were performed in MrBayes v.3.2.2 [37] using the same model as in the ML algorithm. In the BI, the two parallel runs with four chains each (three heated and one cold) were performed for 2 million generations, sampling a tree every 100 generations. The posterior probability values in each node were calculated by FigTree software (version 1.4.3). The consensus tree based on two different trees (achieved from the two methods) was constructed by Dendro Scope software, version 3.2.10 [38]. The average number of nucleotide differences between sampling locations for the full ITS fragment and per nucleotide was estimated in Mega X [39] using the Kimura 2-parameter model [40].

## Population analysis and estimation of relative divergence times

For the population analysis, all sequences of *H. major* were included into the analysis. The number of haplotypes (*N*), haplotype diversity (*h*), and nucleotide diversity ($\pi$) were measured within each region using DnaSP version 6 [41]. Haplotype data were also used to construct a TCS network [42] performed by PopART [43] in order to generate haplotype networks for ITS1-5.8S-ITS2 sequences using their respective alignments. Significant genetic differences among 8 populations ($\Phi_{SC}$), among five regions ($\Phi_{CT}$) (Fig 1), and among individuals ($\Phi_{ST}$) were calculated by non-parametric analysis of molecular variance (AMOVA) to examine the hierarchical population genetic structure by grouping the samples of *H. major* with Arlequin version 3.5 [44].

The relative divergence times of the clades in the *Halophila* spp including *H. ovalis*, *H. major*, *H. minor*, *H. nipponica*, *H. stipulacea*, *H. decipiens* and members of the section *Microhalophila* (*H. beccarii*) were estimated based on the ITS sequence divergence to understand the evolutionary trend of *H. major* using Beast v2.5 [45]. The values between $1.72 \times 10^{-9}$ and $1.71 \times 10^{-8}$ mutations per site and year were used as the range for ITS mutation rates in plants [18, 46]. For Beast analyses, we used a Relaxed Clock Log Normal model. A General Time Reversible (GTR) substitution model with Gamma Categories set to 6 was adopted. The starting tree was randomly generated with a Calibrated Yule process prior. More than 90,000,000 generations of Markov Chain Monte Carlo (MCMC) were implemented of which every 1,000 generations were sampled. The Beast output was analyzed by Tracer v1.7 (Rambaut et al., 2018) [47] and uncertainty in parameter estimates was expressed as values of the 95% highest probability density (HPD). The effective sample sizes of all estimated parameters were also checked in Tracer v1.7 to ensure values were greater than 200. The consensus tree was generated with TreeAnnotator v1.7.3 (Drummond et al., 2012) [48], based on 64,801 trees.

## Results

### *Halophila major* in Vietnamese waters

The leaf shape of *H. major* collected at three different sites, Ly Son Island (Fig 2A), Phu Quoc Island (Fig 2B) and Con Dao Island (Fig 2C) showed a variability, either elliptic or oblong. Among the three populations, leaves collected at Ly Son Island (leaf length = 30.66 ±1 mm; leaf width = 10.12±0.4 mm) were larger than leaves of the two remaining populations (leaf length < 20.0 mm; leaf width < 9.0 mm). However, the number of cross veins of the samples collected at Con Dao (19–22) was higher than those from Ly Son (16–17) and Phu Quoc (14–17). The result also revealed that there were no differences in the branching cross veins between the three populations whereas the space between cross veins of samples collected in Ly Son (1.54 ±0.32 mm) was much wider than those from Con Dao (0.80±0.20 mm) and Phu Quoc (0.90±0.1 mm). Finally, there were no differences in the angle between cross veins and midveins between populations of Ly Son and Phu Quoc (45–60$^0$), but this parameter was higher in Con Dao (75–80$^0$) (Table 1).

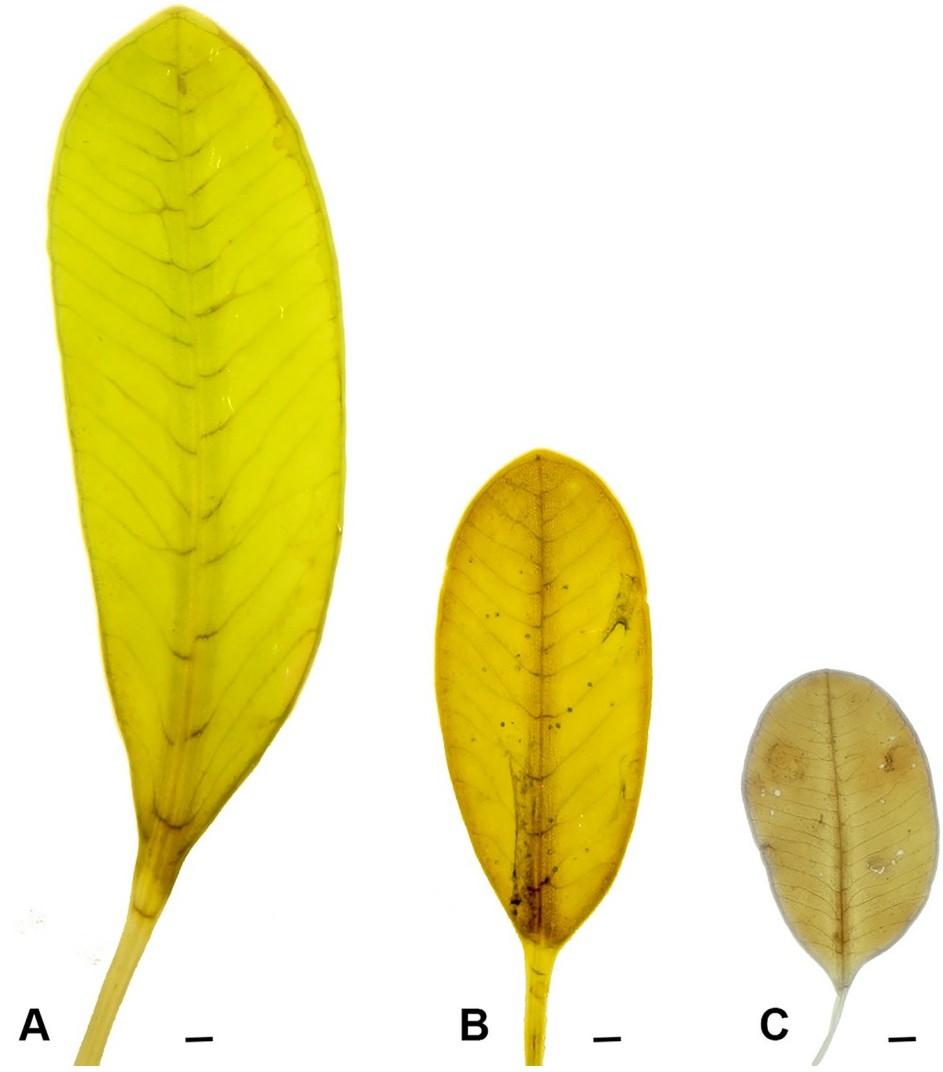

**Fig 2. Variation in morphology of *Halophila major* leaves collected from Vietnamese waters.** A: Ly Son Island; B: Phu Quoc Island; C = Con Dao Island. Scale in each figure = 1 mm.

**Table 1. Leaf dimensions of *Halophila major* collected in the Vietnamese waters.**

| Sites | LW (mm) | LL (mm) | CV (vein) | BCV (vein) | CS (mm) | AG ($^0$) |
|---|---|---|---|---|---|---|
| LS | 10.12±0.36 | 30.66±1.09 | 16–17 | 3–5 | 1.54±0.32 | 45–60 |
| PQ | 8.32±0.30 | 18.10±0.26 | 14–17 | 2–4 | 0.91±0.10 | 45–60 |
| CD | 6.84±0.17 | 11.46±0.11 | 19–22 | 3–5 | 0.80±0.21 | 75–80 |

CV: cross veins, BCV: branching cross veins, SC: space between cross veins, AG: the angle between cross veins and mid-veins, LW: leaf width, and LL: leaf length. See Fig 1 for abbreviations of sampling sites.

A final alignment of 600 bp including gaps was generated for the ITS marker, of which 446 (74.3%) were conserved sites, 142 (23.7%) were variable sites, 59 (9.8%) were parsimony informative characters, and 81 (13.5%) were singletons. Results of the two algorithms applied (ML, BI) showed that all ITS sequences of *H. major* around the world showed eight smaller groups including Sri Lanka (1), Viet Nam/Philippines (2), Japan/Australia (3), Indonesia/Malaysia/Thailand (4) with low support values. Myanmar (5), Australia (6), and the samples of Thailand (7) and Japan (8) also formed the remaining groups (Fig 3). In general, *H. major* groupings are unsupported. Evolutionary divergence as measured by estimated total fragment and per nucleotide differences are 1–16 nucleotides and 0.03–0.28, respectively. In the comparison between samples collected in Viet Nam and other groups, there are 0–2 nucleotide differences among samples collected in Viet Nam and Philippines while the highest number of different nucleotides (14) was found between samples from Viet Nam and Shoalwater, Australia (Table 2).

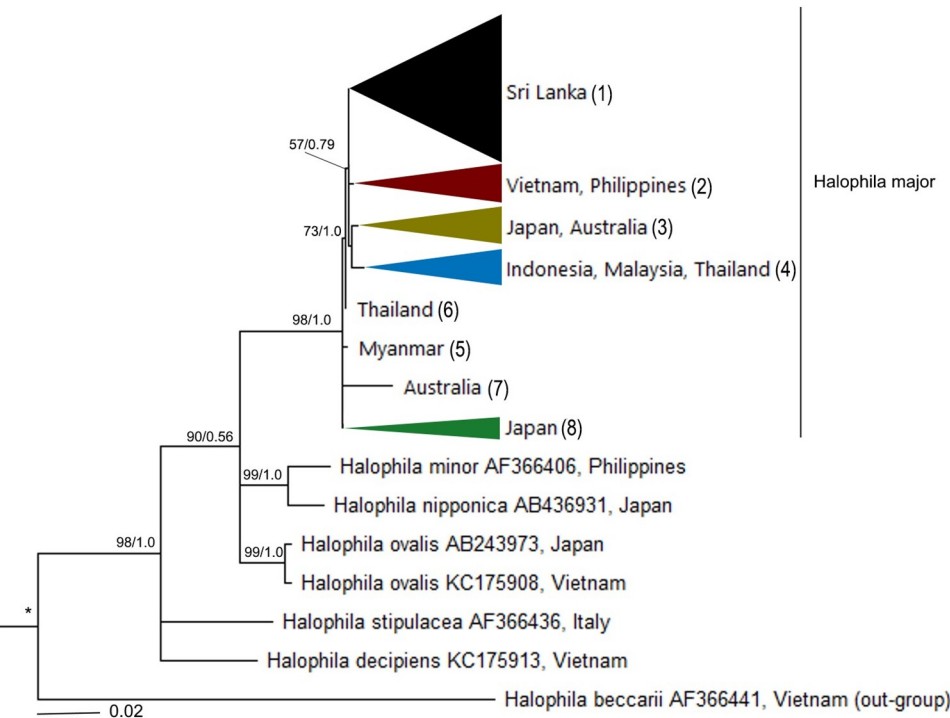

**Fig 3. Phylogenetic tree of members of section *Halophila* inferred from Maximum likelihood and Bayesian inference.** Data set based on 600 bp of ITS1-5.8S-ITS2. Bootstrap values and posterior probability of each method are shown at each node: (left) maximum likelihood; (right) Bayesian inference. *, full support (bootstrap value = 100, posterior probability = 1). 1–8 are smaller groups of *Halophila major*. See S1 Table for more information. The consensus tree was constructed by Dendro Scope software, version 3.2.10.

**Table 2. Evolutionary divergence (un-shading cells) as measured by estimated total fragment and per nucleotide differences (shading cells) of *Halophila major*.**

|   | 1 | 2 | 3 | 4 | 5 | 6 | 7 | 8 |
|---|---|---|---|---|---|---|---|---|
| 1 |  | 0.003 | 0.03–0.012 | 0.05–0.015 | 0.05 | 0.021 | 0.002 | 0.005–0.007 |
| 2 | 2 |  | 0.007–0.015 | 0.009–0.019 | 0.009 | 0.024 | 0.005 | 0.009–0.010 |
| 3 | 2–7 | 4–9 |  | 0.005–0.021 | 0.009–0.014 | 0.019–0.022 | 0.003–0.010 | 0.003–0.012 |
| 4 | 3–9 | 4–11 | 3–12 |  | 0.010–0.021 | 0.019–0.028 | 0.007–0.017 | 0.010–0.022 |
| 5 | 3 | 5 | 4–8 | 6–12 |  | 0.019 | 0.003 | 0.003–0.005 |
| 6 | 12 | 12–14 | 11–12 | 11–16 | 11 |  | 0.019 | 0.015–0.017 |
| 7 | 1 | 3 | 2–6 | 4–10 | 2 | 11 |  | 0.003–0.005 |
| 8 | 3–4 | 5–6 | 2–7 | 6–13 | 2–3 | 9–10 | 2–3 |  |

See Fig 3 for abbreviation of groups.

## Genetic diversity and phylogeography of *Halophila major*

A total of 69 ITS sequences (including three new sequences from the present study) of *H. major* collected in five geographic regions: Sunda Shelf (I), Wallacea (II), Sahul Shelf (III), Bay of Bengal (IV) and coast of Japan (V) generated 22 putative haplotypes (hap01-22). The nucleotide diversity ($\pi$) and haplotype diversity (*Hd*) of ITS within all regions were 0.00458 and 0.710, respectively. Among regions, region II revealed the highest haplotype diversity (1.0) and nucleotide diversity (0.01015) whereas regions I and V showed lower haplotype diversity (0.713 and 0.933, respectively) and nucleotide diversity (0.00478 and 0.00401, respectively). Region IV showed the lowest haplotype diversity (0.202) and nucleotide diversity (0.00035) (Table 3).

A total number of 22 haplotypes (one haplotype from this study and 21 haplotypes deduced from previous studies) showed that seven haplotypes were found in geographic region II (hap01, 06, 08–12) whereas the region I contained five haplotypes (hap01-05). Notably, hap01 was shared by Viet Nam and the Philippines. Hap13-14 were found in the region III only. In the same way, hap15-19 were only distributed in the region V. Among haplotypes in the Bay of Bengal, frequencies of hap20 were highest, as it occupied 89% of the total number. Three haplotypes, hap20-22, were also found in the Bay of Bengal (region IV) (Fig 4). The haplotype network based on the ITS sequences failed to yield some clear phylogeographical separation among the regions (Fig 5). The most parsimonious network revealed two groups comprising two haplotypes, hap05 and hap20. Hap05 (the presumed ancestral haplotype) is at the centre Sunda Shelf and Wallacea (1), and six variants of *H. major* (hap06-12) were raised from hap05.

**Table 3. Summarized *Halophila major* sample size, number of haplotypes observed, and estimates of genetic diversity.**

| Regions | N | h | Hd | $\pi$ | S |
|---------|---|---|----|----|---|
| I | 17 | 7 | 0.713 | 0.00478 | 10 |
| II | 7 | 7 | 1.0 | 0.00583 | 20 |
| III | 2 | 2 | na | na | na |
| IV | 46 | 3 | 0.202 | 0.00035 | 2 |
| V | 6 | 5 | 0.933 | 0.00401 | 5 |
| Over all | 78 | 22 | 0.710 | 0.00458 | 37 |

*N*: Number of sequenced isolates, *h*: number of haplotypes, *Hd*: haplotype diversity, $\pi$: nucleotide diversity, *S*: number of segregating sites. See Fig 1 for abbreviations of regions, na: not available.

* $p < 0.05$.

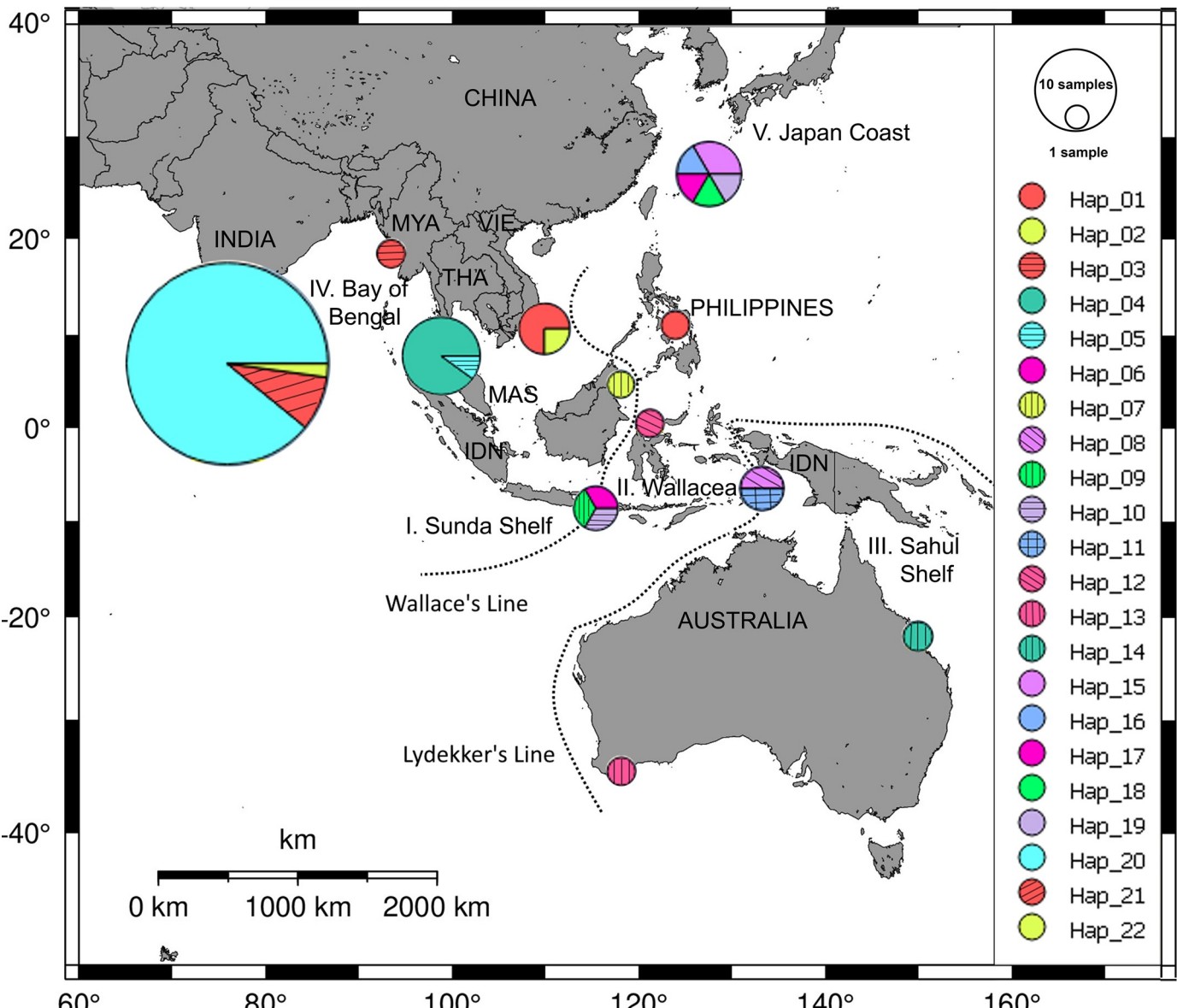

**Fig 4. Distribution of haplotype frequency of *Halophila major* at different regions.** Twenty two haplotypes are defined by different colours/background. The data were processed by PopART software.

There were 1–5 mutations between hap05 and hap06-12. Hap20 may be the central of the remaining groups. In this group, the data trend to form seven smaller groups including Bay of Bengal (1), Viet Nam–the Philippines (2), Japan-Australia (3), Thailand (5), Myanmar (6), Australia (7) and Japan (8). However, there was no clear phylogeographical separation between the coasts of Japan and Australia. The results of AMOVA based on the five regions explained 12.97% of the variation (or fixation index $\varphi_{ST} = 0.8$, *p*-value < 0.01) (Table 4).

## Evolutionary trends and estimation of relative divergence times

The relative divergence times based on ITS1-5.8S-ITS2 for the *Halophila* genus revealed a sequential progression of diversification, from a fairly resolved split between section

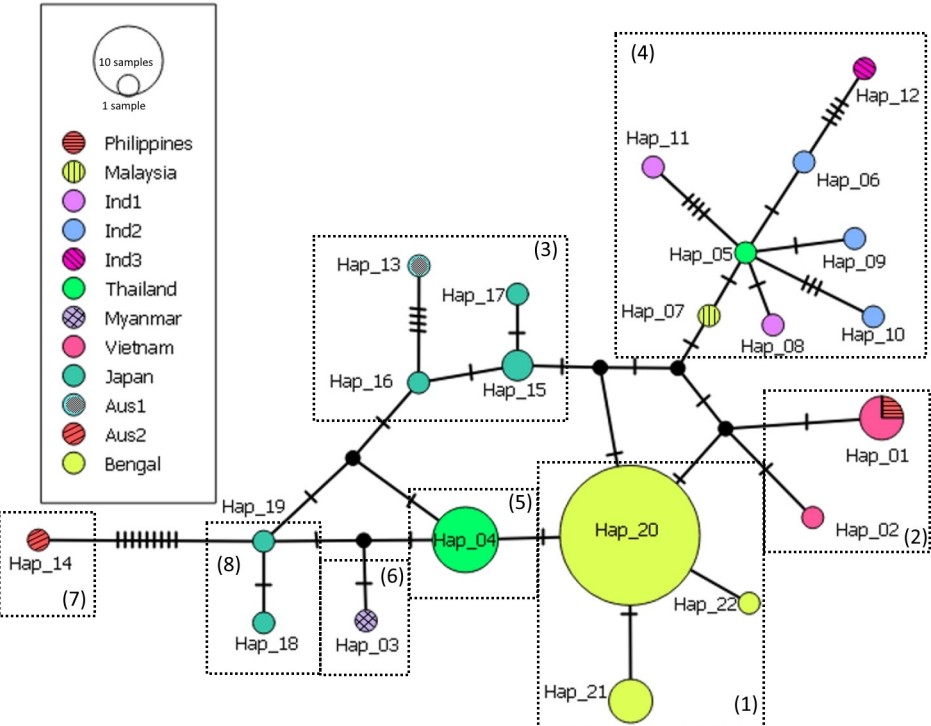

**Fig 5. Haplotype network of 22 haplotypes and their distribution found for *Halophila major* worldwide.**
Haplotypes are written beside or in the circles. Each short segment in the distance between two genotypes is a single mutation. Each dotted line rectangle presents each small group, numbers (1–8) following The data were processed by PopART software.

*Microhalophila* (*H. beccarii*) and the *Halophila* spp (*H. ovalis* and closely related species) (32.18 Mya, posterior probability value, p.p. = 1.0, 95% highest probability density, HPD = 13) to the most recent and highly resolved (p.p. = 1.0) divergence of *H. minor* and *H. nipponica* at 2.15 Mya (95% HPD: 0–4) (Fig 6). Within the section *Halophila*, the relative divergence times for *H. decipiens* and *H. stipulacea* were 11.72 and 10.47 Mya, respectively (p.p. = 1.0, 95% HPD: 7–8). In contrast, the most recent and highly resolved divergences of *H. major*, *H. ovalis*, *H. minor* and *H. nipponica* were 5.44–5.49 Mya (p.p. = 1.0, 95% HPD: 5). Notably, the relative divergence times of *H. ovalis* collected from the Red Sea was 5.44 Mya (p.p = 0.7, 95% HPD: 4). Among the *H. major* groups, the relative divergence times were 1.14–1.71 Mya (p.p. = 1.0, 95% HPD: 1–1.5) (Fig 6).

**Table 4. AMOVA (analysis of molecular variance) results for ITS variation of *Halophila major* collected at five regions.**

| Source of variation | d.f. | SS | $\sigma^2$ | % of variation | Fixation indices |
|---|---|---|---|---|---|
| **Among populations** | 7 | 217.381 | 17.697 | 67.003 | $\Phi_{SC} = 0.769*$ |
| **Among regions** | 4 | 261.376 | 3.425 | 12.966 | $\Phi_{CT} = 0.129$ |
| **Within populations** | 11 | 58.200 | 5.291 | 20.031 | $\Phi_{ST} = 0.799**$ |
| **Total** | 22 | 536.957 | 26.413 | | |

*d.f.*: degree of freedom, *SS*: Sum of squares. See Fig 1 for the regions.

** $p < 0.001$

* $p < 0.05$.

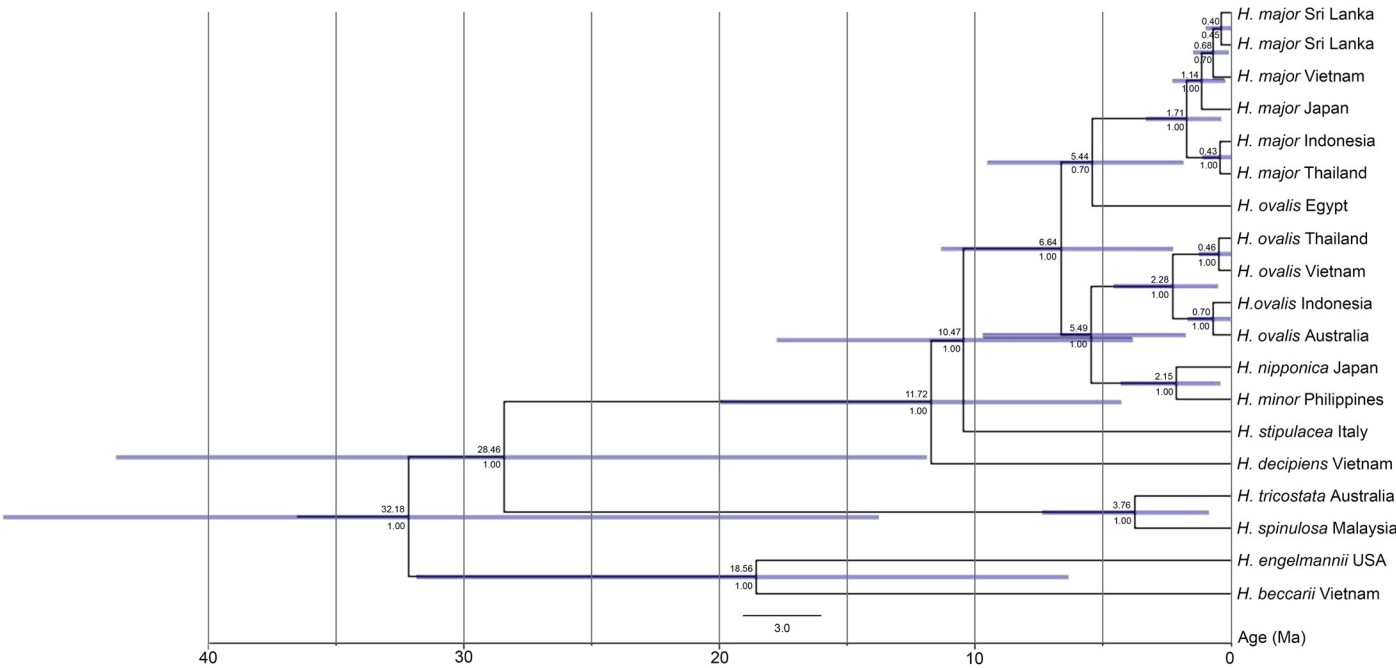

**Fig 6. The relative divergence times based on ITS1-5.8S-ITS2 for *Halophila* spp.** The 95% Highest Density Probability (HPD) intervals are provided at each node; upper value = node divergence time (Mya), lower value = posterior probability values (p.p.). Time-calibrated phylogeny was processed by Beast v2.5.

## Discussion

Our results reveal the high variation of leaf morphology among populations in Vietnamese waters. Kurniawan et al. [10] also indicated that the leaf dimensions of *H. major* collected from different sites in Indonesia also showed variation. The main different morphological character between those from Vietnamese waters and Indonesia is the number of branching cross veins: the number of branching cross veins of samples collected in Viet Nam is significantly lower than in the Indonesian samples (3–4 vs 6–8). The relatively closely related species, *H. ovalis*, showed morphological variability particularly in the leaves (leaf length and leaf width) in response to the different environmental factors in the various habitats [49]. In this present study, the phylogenetic tree based on ITS sequences reveals that all samples collected from the off-shore islands in Viet Nam were *H. major*. This result is in agreement with our previous study in Malaysia using samples collected from Mabul and Gusungan Islands, off the south eastern coast of Sabah [12]. The phylogenetic tree also showed the monophyletic group of the seagrass species *H. major*. Comparing different populations in the Southeast Asian countries and an adjacent region (Japan), the results of Kurniawan et al. [10] indicated two groups of *H. major* whereas samples collected in Malaysia, Indonesia and Thailand tend to form a single group.

The haplotype distribution showed that only hap01 was shared between Viet Nam (region I–Sunda Shelf) and the Philippines (region II) whereas there were no haplotypes shared with other regions. Therefore, boundary lines, such as the Wallace's and Lydekker's Lines, may play an important role as barrier between the Sunda Shelf, Wallacea and the Sahul Shelf, highlighting the strong effects of these geographical barriers also for the evolution of diverse seagrass taxa as was not shown before. Based on this fact of this area as living laboratory of evolution more samples of seagrasses and associated species need to be collected in future studies. A previous study on the haplotype distribution of the sister species *H. ovalis* also indicated that there

are no haplotypes shared among Southeast Asian countries, the Bay of Bengal and the coast of Australia. However, a limited number of haplotypes were shared between Southeast Asian countries and the coast of Japan [19]. For other seagrass species, based on microsatellite analysis, Wainwright et al. [23] found that the manatee grass *Syringodium isoetifolium* formed a cluster that was exclusively located on the shallow Sunda Shelf and appears to follow the demarcation defined by the Wallace's Line. The Wallace's Line is known as the continental margin of the Sunda Shelf, and several other studies on marine snails [50], seahorses [51, 52], and crab [26] have shown similar results observed in our *H. major* study. In contrast, the marine brown alga *Sargassum polycystum* Agardh 1824 showed a homogeneous population throughout Southeast Asia [53]. Haplotype network and distribution of *H. major* showed significant differences between regions separated by the Malay Peninsula which is considered as a geographic barrier for several marine animals and plants. Similar results were also found with *H. ovalis* [12] and with the mangrove species *Lumnitzera racemosa* Willd. 1803 [54]. Within the samples collected from Australia, both phylogenetic analysis and haplotype network revealed two distinct groups and twelve mutations. This may be explained by the long distance between the two sampling sites and different oceanic systems. The average sea temperatures from sampling sites at Japan (between 30–35˚N) and Australia (between 30–35˚S) are similar, around 18–20˚C (**https://phys.org**), and may explain the similarities of *H. major* in two regions.

There were several missing haplotypes, mainly in the Sunda Shelf (Fig 5). Unfortunately, data from the Java Sea, Natuna Archipelago and Singapore Strait were not available. Therefore, it is likely the missing haplotypes may occur in the above mentioned regions. Samples used in this study were stored by dried materials (for voucher specimens) and DESS solution. Fixing samples in DESS solution and store in -20˚C may be the best way for DNA extraction later and morphological observation in the future. Hence, the sub-samples of herbarium voucher specimens should be fixed in DESS solution. More samples in the wider regions should be collected, and international collaboration studies are necessary in order to really get more complete data for better finding out its genetic diversity. The present study also revealed the highest nucleotide diversity of *H. major* in regions I (Sunda Shelf) and II (Wallacea). Previous studies on marine plants also indicated that the highest genetic diversity was also found in Southeast Asian countries, for example, *H. ovalis* [19] and for three species of the mangrove genus *Rhizophora* [55].

The relative divergence times of members of the genus *Halophila* were estimated for the first time based on the ITS marker was estimated. The relative divergence times of *H. major* and *H. ovalis* were similar (6.64 Mya). It is older than what Kim et al. [18] found (around 3.5 Mya). This difference may be based on technical aspects and the length of the sequences in the dataset used. By using multi-loci of the plastid genome, the divergence time estimates between *H. ovalis* and closely related species were 8.4 Mya [17]. Using the single ITS marker also revealed that the divergence time estimates among members of *Halophila* were 2.15 Mya (between *H. minor* and *H. nipponica*) and 11.72 Mya (between *H. decipiens* and remaining species). Moreover, an unexpected result revealed that *H. ovalis* collected form the Red Sea was split from *H. major* instead of *H. ovalis*, and its divergence time estimate was 5.44 Mya. This finding may lead to another hypothesis that *H. ovalis* collected from the Red Sea may be treated as a distinct species or a sub-species of *H. major*. The sequences of plastid genes, for example, *rbc*L, *mat*K and *psb*A-*trn*H from other members of the *Halophila* genus are not available in GenBank. Therefore, our next study will apply multi-locus for the analysis, then the evolution of this genus may be understood in more details.

In conclusion, our findings in this study revealed the haplotype and genetic diversity of *H. major* in Southeast Asian countries and neighbouring regions. *Halophila major* shows

variation in morphology in Viet Nam. Phylogenetic tree showed a monophyletic clade of *H. major*, with unique haplotypes occurring among regions but no or low support for regional groupings. Wallace's and Lydekker's Lines may indicate marine geographic barriers defining to the population structure of *H. major* observed today similar to the patterns seen in many marine organisms, while the Malay Peninsula acts as a geographic land barrier of this seagrass species in the two oceanic systems of the Pacific and Indian Oceans.

## Supporting information

**S1 Table. GenBank accession numbers of the sequences used in the analyses.** [a]: First identi-fied as *Halophila euphlebia*. [b]: used in phylogenetic analysis, [c]: used in population genetic, [d]: used in time-calibrated phylogeny.
(DOCX)

## Acknowledgments

We would like to thank the head of the Institute of Oceanography (ION) for providing the nec-essary facilities to carry out this work. We also thank Mr. Khin Lau of the ION for preparing the map. This paper is a contribution to celebrate the 100 years Anniversary of the Institute of Oceanography, Vietnam Academy of Science and Technology.

## Author Contributions

**Conceptualization:** Xuan-Vy Nguyen, Jutta Papenbrock.

**Data curation:** Xuan-Vy Nguyen, Nhu-Thuy Nguyen-Nhat.

**Formal analysis:** Xuan-Vy Nguyen, Nhu-Thuy Nguyen-Nhat.

**Funding acquisition:** Xuan-Vy Nguyen.

**Investigation:** Xuan-Vy Nguyen, Viet-Ha Dao.

**Methodology:** Xuan-Vy Nguyen, Nhu-Thuy Nguyen-Nhat, Xuan-Thuy Nguyen, Jutta Papenbrock.

**Project administration:** Xuan-Vy Nguyen.

**Resources:** Xuan-Vy Nguyen, Viet-Ha Dao.

**Software:** Xuan-Vy Nguyen, Nhu-Thuy Nguyen-Nhat, Xuan-Thuy Nguyen.

**Supervision:** Jutta Papenbrock.

**Validation:** Xuan-Vy Nguyen, Lawrence M. Liao, Jutta Papenbrock.

**Visualization:** Xuan-Vy Nguyen, Nhu-Thuy Nguyen-Nhat, Xuan-Thuy Nguyen, Viet-Ha Dao, Lawrence M. Liao, Jutta Papenbrock.

**Writing – original draft:** Xuan-Vy Nguyen, Viet-Ha Dao.

**Writing – review & editing:** Xuan-Vy Nguyen, Lawrence M. Liao, Jutta Papenbrock.

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
