## [Decision Letter · Decision Letter 0]

20 Jan 2021

PONE-D-20-31468

Analysis of rDNA reveals a high genetic diversity of Halophila major in the Wallacea region

PLOS ONE

Dear Dr Nguyen, 

Thank you for submitting your manuscript to PLOS ONE. After careful consideration, we feel that it has merit but does not fully meet PLOS ONE’s publication criteria as it currently stands. Therefore, we invite you to submit a revised version of the manuscript that addresses the points raised during the review process. We received the position of the three reviewers and you can see that the opinions are a little dissonant. Anyway, all the reviewers present very relevant considerations and so I chose to indicate a "Major Review". 

We look forward to receiving your revised manuscript.

Kind regards,

Sergio N. Stampar, Dr.

Academic Editor

PLOS ONE

Additional Editor Comments:

Dear Dr Nguyen,

We received the position of the three reviewers and you can see that the opinions are a little dissonant. Anyway, all the reviewers present very relevant considerations and so I chose to indicate a "Major Review". Please address any doubts and issues raised by the reviewers and I look forward to a new version of the manuscript.

2. In your Methods section, please provide additional location information of the sampling sites , including geographic coordinates for the data set if available.

3. In your Methods section, please provide additional information regarding the permits you obtained for the work. Please ensure you have included the full name of the authority that approved the sampling sites  access and, if no permits were required, a brief statement explaining why.

"This work was supported by the National Foundation for Science & Technology

Development, Viet Nam, grant code 106.02-2018.313."

"No, the funders had no role in study design, data collection and analysis, decision to publish, or preparation of the manuscript"

Additionally, because some of your funding information pertains to [commercial funding//patents], we ask you to provide an updated Competing Interests statement, declaring all sources of commercial funding.

In your Competing Interests statement, please confirm that your commercial funding does not alter your adherence to PLOS ONE Editorial policies and criteria by including the following statement: "This does not alter our adherence to PLOS ONE policies on sharing data and materials.” as detailed online in our guide for authors  http://journals.plos.org/plosone/s/competing-interests.  If this statement is not true and your adherence to PLOS policies on sharing data and materials is altered, please explain how.

Please include the updated Competing Interests Statement and Funding Statement in your cover letter. We will change the online submission form on your behalf.

5. We note that Figures 1 and 4 in your submission contain map images which may be copyrighted. All PLOS content is published under the Creative Commons Attribution License (CC BY 4.0), which means that the manuscript, images, and Supporting Information files will be freely available online, and any third party is permitted to access, download, copy, distribute, and use these materials in any way, even commercially, with proper attribution. For these reasons, we cannot publish previously copyrighted maps or satellite images created using proprietary data, such as Google software (Google Maps, Street View, and Earth). For more information, see our copyright guidelines: http://journals.plos.org/plosone/s/licenses-and-copyright.

(1) You may seek permission from the original copyright holder of Figures 1 and 4 to publish the content specifically under the CC BY 4.0 license. 

6. Please include a caption for figures 5 and 6.

Reviewers' comments:

Reviewer's Responses to Questions

**Comments to the Author**

1. Is the manuscript technically sound, and do the data support the conclusions?

Reviewer #1: Yes

Reviewer #2: Partly

Reviewer #3: Partly

2. Has the statistical analysis been performed appropriately and rigorously? 

Reviewer #1: Yes

Reviewer #2: N/A

Reviewer #3: I Don't Know

3. Have the authors made all data underlying the findings in their manuscript fully available?

Reviewer #1: Yes

Reviewer #2: Yes

Reviewer #3: Yes

4. Is the manuscript presented in an intelligible fashion and written in standard English?

Reviewer #1: Yes

Reviewer #2: Yes

Reviewer #3: Yes

5. Review Comments to the Author

Reviewer #1: Ms. Number. : PONE-D-20-31468

Title : Analysis of rDNA reveals a high genetic diversity of Halophila major in the Wallacea region

PLOS ONE

General comments:

The article is very good and very interesting. This article discusses the species diversity of genus Halophila and tries to reveals a high genetic diversity of Halophila major in the Wallacea region using rDNA analysis. This is a good new insight because H. major still has big challenge, especially in Wallacea region. In order to provide a broad impact and strong research and data linkages, it would be better if it could involve cross-country researchers from regional areas, and could be used as a reference for the distribution of Halophila seagrass, especially H. major in ASEAN; I think in Indonesia, this species has just been rediscovered, and I believe it still has a wider range of life, and has a high genetic diversity, not only in the Wallacea region, as you mention in manuscript.

Also, there are several scientific names of species that are not italicized and according to the standard of writing, and should only be the scientific name of the species in italics. And then, there is a lot of writing that needs to be improved, especially related to continuity between sentences and between paragraphs, as well as table and unit. The units should use symbols that have been defined internationally, and are consistent.

Specific comments:

Abstract:

- Clear.

- “The morphological characters show variation, but based on the results of the ITS marker they fom a homogenous population.” Typo ‘fom’?

Introduction:

- Page 3: …“The Halophila section including H. ovalis, H. decipiens Ostenfeld, H. gaudichudii J. Kuo, H. major (Zoll.) Miquel, H. minor (Zollinger) den Hartog, H. nipponica J. Kuo, H. okinawensis J. Kuo and H. stipulacea (Forssk.) Ascherson is known to present as one the most complex challenges plant taxonomy [4,5].” … When biological species that is the first time mentioned in the text, it should attach the authority name and year.

- Page 4: Syringodium isoetifolium, Enhalus acoroides similar comment above.

Material and Methods:

- Page 6: Phylogenetic analysis section….”, including three sequences obtained in this study and 73 sequences of known Halophila sections retrieved from GenBank..”. Total 76 but in Result All sequences were 78.

Results

- Page 7: “Morphological observation and phylogenetic analysis of Halophila major in Vietnamese waters”. Suggestion to reduce, “Halophila major in Vietnamese waters.”

- Page 8: “A total of 78 ITS sequences (including three new sequences from the present study) of H. major collected in five geographic areas: Sunda Shelf (I), Wallacea (II), Sahul Shelf (III), Bay of Bengal (IV) and coast of Japan (V)”.Total sequences are 78 or 76?; this sentence prefer in phylogenetic analysis in method.

- Page 9: in Table 3, please consistent, areas or region is used based on the context.

- Page 10: First paragraph “Area” similar comment above.

Discussion

- Page 11: The number of branching veins for H. ovalis 3-4 from Indonesia samples. Generally, branching cross veins of H. ovalis is 3-4 pairs. How about the branching cross veins of H. ovalis in Vietnam?

- Page 11: “….whereas samples collected in Malaysia, Indonesia and Thailand tend to form a single clade”. Insert Fig. 3 and Why it placed in single clade, How it related with this finding?

- Page 12: Typo. “…..in regions II (Sunda Shelf) and III (Wallacea)”. it means Sunda Shelf (region I) and Wallacea (II)

Figure

- Improve the quality of all Figure (should be min. 350 DPI) follow the journal requirement.

- Figure 1. Suggestion: prefer polygon with different color or pattern to differentiate the region.

- Figure 2. Please insert the scale like 1 cm or 1 mm.

Recommendation: A major revision is required to improve the manuscript.

Reviewer #2: The manuscript entitled "Analysis of rDNA reveals a high genetic diversity of Halophila major in the Wallacea

region" by Nguyen et al. Using only "15" newly collected H. major plants from Viet Nam with other published ITS from GenBank to reveal the phylogentic relationship with other species in the genus Halophila as well the phylogeography of H. major. There are several flaws which are fatal to lead this rejection. First of all, the tile is not falling into their main conclusions(findings), because through out the manuscript, they only discussed about the high genetic diversity in Wallacea with only few sentences in page 12. Second, they seem to be misunderstood the the definition of the term "monophyletic", based on their phylogenetic tree, they mentioned there are 8 subclades in H. major and form nonmonophyletic groups. However, since the sub clades the authors defined in this studied are not well supported at the tips. They is "no" subclade in H. major, and sequences of H. major around the world formed" a "monophyletic clade with well support (supporting values :98/1). This is also supported by the TCS haplotype network which showed no sign of population partition. Third, they use same symbol to code geographic regions as well as 8 clades (I, II, III.....), which is really confusing while reading. Lastly, the resolution of the figures are generally low especially Figure 6.

There are some minor comments as following

Page 5, spell out "DESS"

Figure 1 caption: "where samples were collected change" to "where sequences were obtained from GenBank"

Page 7, "World type" "Red Sea type", you need to explain what are they or simply state previously been defined as "World type"and "Red Sea type" by someone....

There are many sentences developed from the erroneous statement "8 subclades", so the authors may need to reframe the manuscript based on the correct interpretations.

Reviewer #3: The authors analysed rDNA of three H. major collections from Vietnamese islands and similar molecular data from other SE Asia stored in geneBank to demonstrate the origin of this species was located in the Wallacea region. They further estimated the relative divergence times of certain taxa within the section Halophila and also that in section Halophila and section Microhalophila.

Comments:

a. The species Halophila major (H. euphlebia) was ‘re-established’ in 2006, therefore, using the prior to 2006 data from the geneBank to treat as “H. major” could be questionable. For example, the Halophila specimens from Shoalwater Bay, Australia have distinct linear leaf blades with L/W ratio 7-10; cross veins 8-12 (-14), unbranched…Based on the above morphological description, the specimens may not belong to H. major.

b. Abstract stated: “H major was misidentified as H. ovalis in Viet Nam”. However, this statement did not appear in other sections of this manuscript nor any reference was given.

c. Introduction: References [4, 5] did not use section Halophila. Furthermore, H. australis, H. capricorni, and H. sulawesii also include in this section. However, the latest Halophila taxonomy review shows that section Halophila does not include these three species and also not H. stipulacea and H. decipiens (see Kuo 2020).

d. Fig. 3 H. major has eight subclasses. Those from Japan belong to Class III and VIII; Australia belongs to III and VI; while Thailand had IV and VII. Why two different subclasses in the same country (Japan, Australia, Thailand)? Any morphological or environmental difference of the specimens in two different subclasses from the same country?

e. References: should list: Liu et al. (2020) Genetic identification and hybridizationin the seagrass genus Halophila (Hydrocharitaceae) in Sri Lanka waters. PeerJ 2020, 8, e10027

f. Authors estimated relative divergence of H. major and H. ovalis were 6.64 Mya, that was much older than Kim et al.’s estimation of 3.4 Mya. The authors of this manuscript explained these differences were due to ‘technical aspect’ and the length of sequences in the dataset used. They should explain what is ‘technical aspect’.

g. Authors estimated the divergence between section Halophila and section Microhalophila was 32.18 Mya, this estimation implies that the divergence of the genus Halophila would be much earlier than that estimated divergence of genus Halophila in the family Hydrocharitaceae was only 19.41 Mya by Chen et al. The authors of this manuscript should discuss these vast differences.

6. PLOS authors have the option to publish the peer review history of their article (what does this mean?). If published, this will include your full peer review and any attached files.

Reviewer #1: No

Reviewer #2: No

Reviewer #3: No

---

## [Author Response · Author response to Decision Letter 0]

4 Feb 2021

Recommendation Responses

Editor

Q1. Please ensure that your manuscript meets PLOS ONE's style requirements, including those for file naming 

R1: Thank you very much, the new version was formatted following guidelines of journal including the title, authors, affiliations, figure, table and support information. 

Q2. In your Methods section, please provide additional location information of the sampling sites, including geographic coordinates for the data set if available.

R2: We mentioned the additional location information of the sampling sites, including geographic coordinates for the data set in the method section.

Q3. In your Methods section, please provide additional information regarding the permits you obtained for the work. Please ensure you have included the full name of the authority that approved the sampling sites access and, if no permits were required, a brief statement explaining why. 

R3: We added the additional information regarding the permits our team obtained for the work at three locations

Q4. Please remove any funding-related text from the manuscript and let us know how you would like to update your Funding Statement. Currently, your Funding Statement reads as follows

Please include the updated Competing Interests Statement and Funding Statement in your cover letter. We will change the online submission form on your behalf 

R4. Yes, we removed funding-related text from the manuscript. The cover letter is updated.

Q5. We note that Figures 1 and 4 in your submission contain map images which may be copyrighted. All PLOS content is published under the Creative Commons Attribution License (CC BY 4.0), which means that the manuscript, images, and Supporting Information files will be freely available online, and any third party is permitted to access, download, copy, distribute, and use these materials in any way, even commercially, with proper attribution. For these reasons, we cannot publish previously copyrighted maps or satellite images created using proprietary data, such as Google software (Google Maps, Street View, and Earth). For more information, see our copyright guidelines: http://journals.plos.org/plosone/s/licenses-and-copyright. 

R5. The background map used in the manuscript is from NOAA/NGDC. For our previous papers, we contacted the NOAA/NGDC for permission. They answered to us that it is public domain and available for use. Therefore, we wrote “Source of digital map: The National Oceanic and Atmospheric Administration (NOAA), USA, public domain data”

For Fig. 4. The map was integrated in the PopArt software that was used to produce Fig. 4. To be safe, we modified Fig. 4 with background map used in Figure 1.

Q6. Please include a caption for figures 5 and 6. 

R6. Captions for figures 5 and 6 were added

Q7. Please include captions for your Supporting Information files at the end of your manuscript, and update any in-text citations to match accordingly. Please see our Supporting Information guidelines for more information 

R7. Caption for Supporting Information file was included at the end of your manuscript.

We updated citations to match accordingly.

Reviewer 1

Q1. There are several scientific names of species that are not italicized and according to the standard of writing, and should only be the scientific name of the species in italics. And then, there is a lot of writing that needs to be improved, especially related to continuity between sentences and between paragraphs, as well as table and unit. The units should use symbols that have been defined internationally, and are consistent. 

R1. Thank you very much, the manuscript was improved. Please see the changes that were highlighted

Q2. Typos “fom” 

R2. Thank you very much, we corrected

Q3. When biological species that is the first time mentioned in the text, it should attach the authority name and year R3. We added the authority name and year after scientific name

Q4. Page 6: Phylogenetic analysis section….”, including three sequences obtained in this study and 73 sequences of known Halophila sections retrieved from GenBank..”. Total 76 but in Result All sequences were 78 

R4. We changed to …”73 sequences of known Halophila species” 69 ITS sequences of H. major. Please see the S1 Table, 

Q5. Page 7: “Morphological observation and phylogenetic analysis of Halophila major in Vietnamese waters”. Suggestion to reduce, “Halophila major in Vietnamese waters.” 

R5. Thank you very much, we agree to reduce

Q6. Page 8: “A total of 78 ITS sequences (including three new sequences from the present study) of H. major collected in five geographic areas: Sunda Shelf (I), Wallacea (II), Sahul Shelf (III), Bay of Bengal (IV) and coast of Japan (V)”.Total sequences are 78 or 76?; this sentence prefer in phylogenetic analysis in method. 

R6. Yes, we checked and corrected the numbers, please see my answer at R4. 

Q7. Page 9: in Table 3, please consistent, areas or region is used based on the context 

R7. Thank you for your finding, we changed all “area” to “region” except where area was used for specific locations in Viet Nam

Q8. Page 10: First paragraph “Area” similar comment above. 

R8. Please, see our answer in R7

Q9. Page 11: The number of branching veins for H. ovalis 3-4 from Indonesia samples. Generally, branching cross veins of H. ovalis is 3-4 pairs. How about the branching cross veins of H. ovalis in Vietnam? 

R9. Thank you very much for your sharing, it is the same, 3-4 from the Vietnamese samples

Q10. Page 11: “….whereas samples collected in Malaysia, Indonesia and Thailand tend to form a single clade”. Insert Fig. 3 and Why it placed in single clade, How it related with this finding? 

R10. We inserted Fig 3. The group IV contains materials from Malaysia, Indonesia and Thailand were performed by data. There are several articles mentioning long‐distance dispersal, positively buoyant shoots with attached rhizomes or seedlings having high potential for long‐distance dispersal. Therefore, DNA fingerprinting should be applied to investigate the population structure and gene flows among populations of Halophila ovalis or Halophila major from SEA in further research studies. 

Q11. Page 12: Typo. “…..in regions II (Sunda Shelf) and III (Wallacea)”. it means Sunda Shelf (region I) and Wallacea (II) R11. Thank you very much, we corrected

Q12. Improve the quality of all Figure (should be min. 350 DPI) follow the journal requirement. 

R12. We agree, all figures were improved

Q13. Figure 1. Suggestion: prefer polygon with different color or pattern to differentiate the region. 

R13. Thank you for your suggestion. We tried polygon, but the land is polygon too. However, Figure 1 was modified following reviewer 2 by using different symbols coding different regions

Q14. Figure 2. Please insert the scale like 1 cm or 1 mm 

R14. We agree, we edited the figure 2, the scale is 1 mm that was mentioned in figure caption.

Reviewer 2

Q1: the title is not falling into their main conclusions (findings), because throughout the manuscript, they only discussed about the high genetic diversity in Wallacea with only few sentences in page 12. 

R1. We add more sentences, it may support to the tittle, please see it in the highlight.

Q2. Second, they seem to be misunderstood the the definition of the term "monophyletic", based on their phylogenetic tree, they mentioned there are 8 subclades in H. major and form nonmonophyletic groups. However, since the sub clades the authors defined in this studied are not well supported at the tips. They is "no" subclade in H. major, and sequences of H. major around the world formed" a "monophyletic clade with well support (supporting values: 98/1). This is also supported by the TCS haplotype network which showed no sign of population partition. 

R2. Thank you very much for your suggestion, we changed to smaller groups, and statement “and sequences of H. major around the world formed" a "monophyletic clade”

Q3. Third, they use same symbol to code geographic regions as well as 8 clades (I, II, III.....), which is really confusing while reading 

R3. We changed different symbols to different regions in Fig. 1. Eight small groups were also modified

Q4. the resolution of the figures are generally low especially Figure 6. 

R4. Thank you, we improved the resolution all figure

Q5. Page 5, spell out "DESS" 

R5. We explained DESS solution. It is 20% dimethyl sulfoxide, 0.25 M disodium EDTA, and saturated NaCl.

Q6 Figure 1 caption: "where samples were collected change" to "where sequences were obtained from GenBank" 

R6. Thank you very much. We changed, and modified the Figure 1 caption. Please see our answer R13 to reviewer 1.

Q7. "World type" "Red Sea type", you need to explain what are they or simply state previously been defined as "World type"and "Red Sea type" by someone....

There are many sentences developed from the erroneous statement "8 subclades", so the authors may need to reframe the manuscript based on the correct interpretations. 

R7. We corrected and removed Red Sea type, simply we changed to “samples collected from Red Sea”, and we changed “subclade” to “group”

Reviewer 3

Q1. The species Halophila major (H. euphlebia) was ‘re-established’ in 2006, therefore, using the prior to 2006 data from the geneBank to treat as “H. major” could be questionable. For example, the Halophila specimens from Shoalwater Bay, Australia have distinct linear leaf blades with L/W ratio 7-10; cross veins 8-12 (-14), unbranched…Based on the above morphological description, the specimens may not belong to H. major. 

R1. Thank you very much, we agree with you. It is not easy to identify the Halophila major and closely related species based on morphological observation only. In this present study, first we used the sequence of H. major (Uchimura et al. 2008) for blasting in NCBI. All sequences with similarity more than 99% were used for the analysis. This group contained two sequences from Australia. Intra-species variation within H. nipponica were reported in Japan or H. ovalis from Malaysia.

Q2. Abstract stated: “H major was misidentified as H. ovalis in Viet Nam”. However, this statement did not appear in other sections of this manuscript nor any reference was given. 

R2. The Halophila ovalis/major complex was reported from Viet Nam in 2013 (Ref 13), and H. major was only found at one off-shore island. Therefore, we are now checking the Halophila ovalis/major complex from three different locations. 

Q3. Introduction: References [4, 5] did not use section Halophila. Furthermore, H. australis, H. capricorni, and H. sulawesii also include in this section. However, the latest Halophila taxonomy review shows that section Halophila does not include these three species and also not H. stipulacea and H. decipiens (see Kuo 2020). 

R3. We modified these sentences, 13 species in section Halophila was updated, and the Ref. Kuo, 2020 was updated. Please see our answer to reviewer 1.

Q4. Fig. 3 H. major has eight subclasses. Those from Japan belong to Class III and VIII; Australia belongs to III and VI; while Thailand had IV and VII. Why two different subclasses in the same country (Japan, Australia, Thailand)? Any morphological or environmental difference of the specimens in two different subclasses from the same country? 

R4. It is a really nice question. Your question is something that is related to Q11 from reviewer 1. In Australia, I think that the geographic distance between two sites from Australia, it may be about 3,000 km from the west side to the east site. In the case of Japan, one site is Okinawa where the water temperature is warmer than the mainland (the other site), but we cannot explain in the case of Thailand. Two haplotypes were found in the Andaman Sea. Hopefully, we will collect more materials from above locations, then we use SSRs approach to estimate the migration among populations in SEA and the other regions 

Q5. References: should list: Liu et al. (2020) Genetic identification and hybridizationin the seagrass genus Halophila (Hydrocharitaceae) in Sri Lanka waters. PeerJ 2020, 8, e10027 

R5. Thank you, we cited and added this Ref.

Q6. Authors estimated relative divergence of H. major and H. ovalis were 6.64 Mya, that was much older than Kim et al.’s estimation of 3.4 Mya. The authors of this manuscript explained these differences were due to ‘technical aspect’ and the length of sequences in the dataset used. They should explain what is ‘technical aspect’. 

R6. Thank you for your interesting comment. The different estimated relative divergence in Kim et al. 2017 can be explained as follows: Figure 4A (estimated relative divergence = 3.5 Mya for H. ovalis), and Figure 4B (estimated relative divergence is about 8 Mya for H. ovalis clade). The software applied in their work was NETWORK 4.6 program. Therefore. the 95% Highest Density Probability (HPD) intervals and posterior probability values were not included. In this present study we used the popular software for estimating relative divergences Beast v2.5

Q7 Authors estimated the divergence between section Halophila and section Microhalophila was 32.18 Mya, this estimation implies that the divergence of the genus Halophila would be much earlier than that estimated divergence of genus Halophila in the family Hydrocharitaceae was only 19.41 Mya by Chen et al. The authors of this manuscript should discuss these vast differences. 

R7. We agree with you. Chen et al used multilocus in plastid DNA while this present study used rDNA. Unfortunately, there are only limited data of plastid DNA from members of Halophila.

---

## [Decision Letter · Decision Letter 1]

19 Apr 2021

PONE-D-20-31468R1

Analysis of rDNA reveals a high genetic diversity of Halophila major in the Wallacea region

PLOS ONE

Dear Dr. Nguyen

Thank you for submitting your manuscript to PLOS ONE. After careful consideration, we feel that it has merit but does not fully meet PLOS ONE’s publication criteria as it currently stands. Therefore, we invite you to submit a revised version of the manuscript that addresses the points raised during the review process.

Thank you very much for the new version of the manuscript. Note that one of the reviewers still expects a few changes before his manuscript is accepted for publication. Please make these adjustments so that I can proceed with the revision of the manuscript.

We look forward to receiving your revised manuscript.

Kind regards,

Sergio N. Stampar, Dr.

Academic Editor

PLOS ONE

Journal Requirements:

Additional Editor Comments (if provided):

Dear authors,

Thank you very much for the new version of the manuscript. Note that one of the reviewers still expects a few changes before his manuscript is accepted for publication. Please make these adjustments so that I can proceed with the revision of the manuscript.

Kind regards

Sergio

Reviewers' comments:

Reviewer's Responses to Questions

**Comments to the Author**

1. If the authors have adequately addressed your comments raised in a previous round of review and you feel that this manuscript is now acceptable for publication, you may indicate that here to bypass the “Comments to the Author” section, enter your conflict of interest statement in the “Confidential to Editor” section, and submit your "Accept" recommendation.

Reviewer #1: All comments have been addressed

Reviewer #2: (No Response)

2. Is the manuscript technically sound, and do the data support the conclusions?

Reviewer #1: Yes

Reviewer #2: Partly

3. Has the statistical analysis been performed appropriately and rigorously? 

Reviewer #1: Yes

Reviewer #2: Yes

4. Have the authors made all data underlying the findings in their manuscript fully available?

Reviewer #1: Yes

Reviewer #2: Yes

5. Is the manuscript presented in an intelligible fashion and written in standard English?

Reviewer #1: Yes

Reviewer #2: Yes

6. Review Comments to the Author

Reviewer #1: In the discussion section, suggestions should be made regarding the use of fresh specimens and adding locations, and things that become input and challenges in the future, including collaborative research in order to really get more complete data to better find out its genetic diversity.

Also, there is something that needs to be ascertained from the phylogenetic tree, is H. major in Australia really any specimens based on morphological evidence? If there is, it may be stated that the source of the citation is that H. major is found in Australia in Materials and methods.

Reviewer #2: Although authors have replied most of my comments, there is one exception which related to the symbol (I, II, III etc.) they used in the Figure 1 and in the phylogenetic tree to represent geographic areas and "groups", respectively.

They are several things that I concern while reviewing this revision. Here I list them point by point as follows.

Line 75-76: Delete "In addition, the study....on the IST marker."

Line 88-89: Unclear, had to rephrase.

Line 106-108: This sentence is odd, need to rewrite to clarify what you were trying to say.

Line 109-110:How the genetic partition occur "within" and "among" barrier ? The authors may need to clarify this or simply make this sentence more straight forward.

Line 129: Use "sequences of other regions" instead of "other regions sequences"

Line 140: Use "the" instead of "this".

Line 243-244:I would like the authors to provide the supporting values of each groups they defined. Because technically, if those groups are not support by supporting values they should not be defined as different group. I know the authors already used group instead of clade. One thing the authors can do is be honest with what they found (groups are not well-supported) and add up the information that the grouping of those haplotypes might affiliate with geographic location based on the haplotype network instead of define them as groups without substantial evidence.

Line 328-329:The authors should explain what is the "similar way" they mentioned here? Because the original sentence is not very clear. Additionally, in the tree generated by Waycott et al. 2002, those groups found in H. ovalis samples were not well-supported, therefore, they also over-interpreted their data almost 20 years ago. And I truly believe we should not follow the conclusion which is not correct.

Line 382-383:You only have the morphological data from Viet Nam, therefore, I don't think you can make this conclusion while you mentioned a broad area in previous sentence.

I strongly recommend the authors send their next revision for English editing before submitting back to PloS One, because I did not correct them across the manuscript. I am not a native speaker too, and I always send my manuscript for English editing before submitting, just because it actually help the audiences to read and receive the information that we are trying to deliver.

7. PLOS authors have the option to publish the peer review history of their article (what does this mean?). If published, this will include your full peer review and any attached files.

Reviewer #1: **Yes: **Fery Kurniawan

Reviewer #2: No

---

## [Author Response · Author response to Decision Letter 1]

25 Apr 2021

Dear Editor

Thank you very much for forwarding the detailed suggestions and comments from the reviewers. All suggestions have been very helpful for us to improve the manuscript. All changes are highlighted in the newest version. Our responses are below:

List of revisions

Editor

Q1. Note that one of the reviewers still expects a few changes before his manuscript is accepted for publication. Please make these adjustments so that I can proceed with the revision of the manuscript. 

R1: Thank you very much, please see our responses to the reviewers below.

Reviewer 1

Q1. In the discussion section, suggestions should be made regarding the use of fresh specimens and adding locations, and things that become input and challenges in the future, including collaborative research in order to really get more complete data to better find out its genetic diversity. 

R1. Thank you very much for your suggestions, we add more information in discussion part (yellow highlight). 

Q2. There is something that needs to be ascertained from the phylogenetic tree, is H. major in Australia really any specimens based on morphological evidence? If there is, it may be stated that the source of the citation is that H. major is found in Australia in Materials and methods.

R2. We agree with you, “H. major” was previously treated as H. ovalis by Waycott et al. (2002). Uchimura et al. (2008) suggested that it should be H. major. Therefore, we add Uchimura et al. (2008) and Waycott et al. (2002) when we mention that H. major occurs in Australia (yellow highlight. Line 63, page 3)

Reviewer 2

Q1: Although authors have replied most of my comments, there is one exception which related to the symbol (I, II, III etc.) they used in the Figure 1 and in the phylogenetic tree to represent geographic areas and "groups", respectively. R1. Thank you very much for your finding the conflicts between Fig. 1 and Fig. 3. We kept I, II, III in Fig. 1. We changed 1,2,3… in Figure 3

Q2 They are several things that I concern while reviewing this revision. Here I list them point by point as follows.

Line 75-76: Delete "In addition, the study....on the IST marker."

Line 88-89: Unclear, had to rephrase.

Line 106-108: This sentence is odd, need to rewrite to clarify what you were trying to say.

Line 109-110:How the genetic partition occur "within" and "among" barrier ? The authors may need to clarify this or simply make this sentence more straight forward.

Line 129: Use "sequences of other regions" instead of "other regions sequences"

Line 140: Use "the" instead of "this". 

R2. Thank you for your recommendation

Line 75-76: Deleted 

Line 88-89: Yes, we modified the sentence.

Line 106-108: Thank you, we rewrote the sentence.

Line 109-110: We changed this sentence.

Line 129: Yes, we modified.

Line 140: done

Q3. Line 243-244: I would like the authors to provide the supporting values of each groups they defined. Because technically, if those groups are not support by supporting values they should not be defined as different group. I know the authors already used group instead of clade. One thing the authors can do is be honest with what they found (groups are not well-supported) and add up the information that the grouping of those haplotypes might affiliate with geographic location based on the haplotype network instead of define them as groups without substantial evidence. 

R3. Thank you, we agree with you, we added the supporting values to the tree. Therefore, the Fig. 3 was modified. We agree with you, H. major is a monophyletic group, but the trend seems to separate it into groups with low supporting values (not for all groups). You can follow our changes by highlighting them. For the haplotype network, we also agree with you and modified Fig. 5. We removed group A,B. The text was also modified, please see the highlighted passages. 

Q4. Line 328-329: The authors should explain what is the "similar way" they mentioned here? Because the original sentence is not very clear. Additionally, in the tree generated by Waycott et al. 2002, those groups found in H. ovalis samples were not well-supported, therefore, they also over-interpreted their data almost 20 years ago. And I truly believe we should not follow the conclusion which is not correct. 

R4. We removed two last sentences of this paragraph.

Q5. Line 382-383: You only have the morphological data from Viet Nam, therefore, I don't think you can make this conclusion while you mentioned a broad area in previous sentence. 

R5. We changed to monophyletic group that was not changed in the previous version.

Q6 I strongly recommend the authors send their next revision for English editing before submitting back to PloS One, because I did not correct them across the manuscript. I am not a native speaker too, and I always send my manuscript for English editing before submitting, just because it actually help the audiences to read and receive the information that we are trying to deliver. 

R6. Thank you for your recommendation, we will consider your suggestion. Our co-authors, especially Prof. Lawrence M. Liao and Prof. J. Papenbrock carefully checked and edited the manuscript in the final version.

We hopefully fulfilled all points raised.

Sincerely yours,

Xuan-Vy Nguyen

---

## [Decision Letter · Decision Letter 2]

28 Jul 2021

PONE-D-20-31468R2

Analysis of rDNA reveals a high genetic diversity of Halophila major in the Wallacea region

PLOS ONE

Dear Dr. Nguyen,

Thank you for submitting your manuscript to PLOS ONE. After careful consideration, we feel that it has merit but does not fully meet PLOS ONE’s publication criteria as it currently stands. Therefore, we invite you to submit a revised version of the manuscript that addresses the points raised below.

The revised manuscript is well written and reviewer 1 has indicated that their prior comments have all been met. As the stand-in Subject Editor for the manuscript, I appreciate the authors’ and prior reviewers’ efforts to ensure the accuracy and clarity of the manuscript. However, I feel there are several issues that need to be addressed prior to publication.

Principally, results as presented do not clearly support the primary conclusion of significant genetic isolation of *Halophila major* within the central Indo-Pacific (lines 42-43 and elsewhere). I base this specifically on the lack of phylogenetic support for putative *H. major* clades and the non-significant PhiCT values presented in the AMOVA table (table 4).

The lack of support for phylogenetic divergence within H. major is acknowledged in the text (eg. lines 381-383) yet elsewhere support for the putative groupings is implied (eg. lines 38-39). At a minimum, this confuses the discussion and should be clarified.  

As described in Excoffier et al. 2010, AMOVA allows the hierarchical partitioning of genetic variation among populations (phiSC), among regions (phiCT), and among individuals within populations (phiST). As presented, table 4 indicates a non-significant phiCT. This may simply be a typo, as suggested by the lack of haplotype sharing among sites (table 2). If, however, this is not a typo, I feel the primary conclusions of the manuscript are unsupported and significant revisions would be needed before publication.

I am therefore recommending major revisions with the caveat that my concerns about project findings may be addressed if AMOVA results in table 4 are not correct as presented.

Below I describe additional recommendations and concerns that should be addressed.

Lines 38-39: Misidentification of Viet Nam H. major was described in a prior study and should not be highlighted here as a new finding.

Line 40: It is unclear who “they” is referencing. Viet Nam samples only?

Lines 42-43: If the AMOVA table is correct as presented this conclusion will need to be revised

Line 52: I’m unfamiliar with using “sections” to describe phylogenetic clades. Is that common in plants? Please correct if not.

Lines 54-59: Author taxonomy citations are inconsistent. My understanding is that presenting the authors last name is standard format. Please confirm and correct as needed.

Line 104-105: Sentence starting with “Oceanic currents…” is unclear. Please revise.

Line 118: Delete “area”

Line 146: Change “Information of the samples…” to “Sample information…”

Methods: Significant methodological details are missing…

How were pairwise nucleotide differences estimated? What mutational model was used?What is a p-distance?How were AMOVA groupings determined and what were they?Highest Probability Density (BEAST?) is not described.

Lines 231-232: The described H. major groupings are unsupported and should be collapsed into a single branch. Alternatively, the text should be clarified here and elsewhere to avoid any suggestion that there is phylogenetic support for the groups. This does not alone refute the project findings, but it does add some uncertainty.

Table 3: Type “summared”

Lines 281-282: The AMOVA statement will need to be revised if table 4 is correct.

Line 309: Typo “highest density probability”

We look forward to receiving your revised manuscript.

Kind regards,

Jeffrey A. Eble, Ph.D.

Academic Editor

PLOS ONE

Reviewers' comments:

Reviewer's Responses to Questions

**Comments to the Author**

1. If the authors have adequately addressed your comments raised in a previous round of review and you feel that this manuscript is now acceptable for publication, you may indicate that here to bypass the “Comments to the Author” section, enter your conflict of interest statement in the “Confidential to Editor” section, and submit your "Accept" recommendation.

Reviewer #1: All comments have been addressed

2. Is the manuscript technically sound, and do the data support the conclusions?

Reviewer #1: Yes

3. Has the statistical analysis been performed appropriately and rigorously? 

Reviewer #1: Yes

4. Have the authors made all data underlying the findings in their manuscript fully available?

Reviewer #1: Yes

5. Is the manuscript presented in an intelligible fashion and written in standard English?

Reviewer #1: Yes

6. Review Comments to the Author

Reviewer #1: (No Response)

7. PLOS authors have the option to publish the peer review history of their article (what does this mean?). If published, this will include your full peer review and any attached files.

Reviewer #1: No

---

## [Author Response · Author response to Decision Letter 2]

11 Aug 2021

Dear Editor

Thank you very much for your detailed suggestions and comments. All your suggestions are very helpful for us to improve the manuscript. All changes are highlighted in the newest version. Our responses are below:

List of revisions

Recommendation Responses

Editor

Q1. Principally, results as presented do not clearly support the primary conclusion of significant genetic isolation of Halophila major within the central Indo-Pacific (lines 42-43 and elsewhere) 

R1: We agree with you, we changed to non-significant genetic isolation.

Q2. The lack of support for phylogenetic divergence within H. major is acknowledged in the text (eg. lines 381-383) yet elsewhere support for the putative groupings is implied (eg. lines 38-39). At a minimum, this confuses the discussion and should be clarified.

R2: We modified lines 38-39, and line 381-383

Q3. As described in Excoffier et al. 2010, AMOVA allows the hierarchical partitioning of genetic variation among populations (phiSC), among regions (phiCT), and among individuals within populations (phiST). As presented, table 4 indicates a non-significant phiCT. This may simply be a typo, as suggested by the lack of haplotype sharing among sites (table 2). If, however, this is not a typo, I feel the primary conclusions of the manuscript are unsupported and significant revisions would be needed before publication. 

R3. We agree with you, it is typos, Table 4 was corrected

Q4. I am therefore recommending major revisions with the caveat that my concerns about project findings may be addressed if AMOVA results in table 4 are not correct as presented. 

R4. We changed the results of Table 4. Non-significant differences were found among regions, but significant differences were presented among populations

Q5. Lines 38-39: Misidentification of Viet Nam H. major was described in a prior study and should not be highlighted here as a new finding 

R.5 It was removed

Q6. Line 40: It is unclear who “they” is referencing. Viet Nam samples only? 

R6. We modified the sentence.

Q7. Lines 42-43: If the AMOVA table is correct as presented this conclusion will need to be revised 

R7. Please see our responses in R5 and R6

Q8. Lines 54-59: Author taxonomy citations are inconsistent. My understanding is that presenting the authors last name is standard format. Please confirm and correct as needed. 

R8. We used authors last name, and they are consistent now

Q9. Line 104-105: Sentence starting with “Oceanic currents…” is unclear. Please revise. 

R9. We used “sea currents”

Q10. Line 118: Delete “area” 

R10. Done

Q11. Line 146: Change “Information of the samples…” to “Sample information…” 

R11. Done

Q12. Methods: Significant methodological details are missing…How were pairwise nucleotide differences estimated? What mutational model was used? What is a p-distance? How were AMOVA groupings determined and what were they? Highest Probability Density (BEAST?) is not described. 

R12. We added significant methodological details.

Q13. Lines 231-232: The described H. major groupings are unsupported and should be collapsed into a single branch. Alternatively, the text should be clarified here and elsewhere to avoid any suggestion that there is phylogenetic support for the groups. This does not alone refute the project findings, but it does add some uncertainty. 

R13. Thank you very much, our changes to mention that the H. major groupings are unsupported.

Q14. Table 3: Type “summared” 

R14. Done

Q15. Lines 281-282: The AMOVA statement will need to be revised if table 4 is correct. 

R15. Done

Q16. Line 309: Typo “highest density probability” 

R16. Done

We hopefully fulfilled all points raised.

Sincerely Yours

Xuan-Vy Nguyen

---

## [Editor Report · Decision Letter 3]

23 Aug 2021

PONE-D-20-31468R3

Analysis of rDNA reveals a high genetic diversity of Halophila major in the Wallacea region

PLOS ONE

Dear Dr. Nguyen,

Thank you for submitting your revised manuscript to PLOS ONE. After careful consideration, we feel that you have successfully addressed comments on the previous draft manuscript (revision 2) and no additional changes are requested aside from grammar and wording issues listed below. We invite you to submit a revised version of the manuscript after addressing the issues listed below.

Lines 37-38: "show variation" is unclear. For clarity I recommend indicating among what groups variation was observed. With the addition of this information I as well recommend splitting this statement into two sentences, the first highlighting where variation is observed and the second indicating low support for regional H. major groupings.

Lines 105-106. Sorry for the confusion. My prior concerns about this sentence were not related to the term 'oceanic currents', rather I feel your statement that currents can "act as a gene-exchange line for the mirgration requiring long-distance dispersal abilities" is unclear. I'm pretty sure I understand your point, perhaps something like "Oceanic currents can act to both promote or limit gene-exchange" would be more clear.

Figure 1: The Myanmar  site currently shows a solid circle but instead should be a solid diamond based on the figure 1 legend.

Line 146: "Sample" not "Samples"

Lines 191-193: Suggested revision "The average number of nucleotide differences between sampling locations for the full ITS fragment and per nucleotide was estimated in Mega X (40) using the Kimura 2-parameter model (39)."

Lines 249-250: Which ML and base values are you referring to? I would not consider the values in general to be high given the lack of support for the H. major group. Please revise to avoid giving the false impression that there is significant support for H. major groupings.

Line 251 and elsewhere: Table 2 presents two different estimates of 'evolutionary divergence', one for the full fragment and one standardized estimate of the per nucleotide rate. I recommend changing how these measures are referred to here and elsewhere for clarity. Perhaps something like "Evolutionary divergence as measured by estimated total fragment and per nucleotide differences...".

Table 2 caption: See my note above.

Table 3 caption: "Summarized statistics of haplotypes" is unclear. I recommend something like "Summarized Halophila major sample size, number of haplotypes observed, and estimates of genetic diversity." 

Table 3: I only just noticed that you've included estimates of Tajima's D and Fu's Fs here but they are not mentioned in the results nor discussion. I recommend deleting these measures from table 3 and the methods unless you can find a meaningful way to include the results.

Line 401: Recommend changing "but it trends to form..." to "with unique haplotypes occurring among regions but no or low support for regional groupings."

We look forward to receiving your revised manuscript.

Kind regards,

Jeffrey A. Eble, Ph.D.

Academic Editor

PLOS ONE
---

## [Author Response · Author response to Decision Letter 3]

24 Aug 2021

List of revisions

Recommendation Responses

Editor

Q1. Lines 37-38 "show variation" is unclear. For clarity I recommend indicating among what groups variation was observed. With the addition of this information I as well recommend splitting this statement into two sentences, the first highlighting where variation is observed and the second indicating low support for regional H. major groupings 

R1: We agree with you, we changed by splitting into two sentences.

Q2. Lines 105-106.. perhaps something like "Oceanic currents can act to both promote or limit gene-exchange" would be more clear.

R2: Thank you for your suggestion. We changed

Q3. Figure 1: The Myanmar site currently shows a solid circle but instead should be a solid diamond based on the figure 1 legend.

R3. I think it should by a solid circle, please see our highlight on the figure 1 legend

Q4. Line 146: "Sample" not "Samples"

R4. Done

Q5. Lines 191-193: Suggested revision "The average number of nucleotide differences between sampling locations for the full ITS fragment and per nucleotide was estimated in Mega X (40) using the Kimura 2-parameter model (39).

R.5 Thank you for your correction, we changes. We also changed in the Reference. Please see our highlight

Q6. Lines 249-250: Which ML and base values are you referring to? I would not consider the values in general to be high given the lack of support for the H. major group. Please revise to avoid giving the false impression that there is significant support for H. major groupings. 

R6. We removed the sentence.

Q7. Line 251 and elsewhere: Table 2 presents two different estimates of 'evolutionary divergence', one for the full fragment and one standardized estimate of the per nucleotide rate. I recommend changing how these measures are referred to here and elsewhere for clarity. Perhaps something like "Evolutionary divergence as measured by estimated total fragment and per nucleotide differences...".

R7. We changed

Q8. Table 2 caption: See my note above.. 

R8. We changed

Q9. Table 3 caption: "Summarized statistics of haplotypes" is unclear. I recommend something like "Summarized Halophila major sample size, number of haplotypes observed, and estimates of genetic diversity 

R9. Done

Q10. Table 3: I only just noticed that you've included estimates of Tajima's D and Fu's Fs here but they are not mentioned in the results nor discussion. I recommend deleting these measures from table 3 and the methods unless you can find a meaningful way to include the results 

R10. We removed in both Table and in Method

Q11. Line 401: Recommend changing "but it trends to form..." to "with unique haplotypes occurring among regions but no or low support for regional groupings."

R11. Thank you, we changed

---

## [Editor Report · Decision Letter 4]

11 Oct 2021

Analysis of rDNA reveals a high genetic diversity of Halophila major in the Wallacea region

PONE-D-20-31468R4

Dear Dr. Nguyen,

We’re pleased to inform you that your manuscript has been judged scientifically suitable for publication and will be formally accepted for publication once it meets all outstanding technical requirements.

Kind regards,

Jeffrey A. Eble, Ph.D.

Academic Editor

PLOS ONE
---

## [Editor Report · Acceptance letter]

13 Oct 2021

PONE-D-20-31468R4 

Analysis of rDNA reveals a high genetic diversity of *Halophila major* in the Wallacea region 

Dear Dr. Nguyen:

I'm pleased to inform you that your manuscript has been deemed suitable for publication in PLOS ONE. Congratulations! Your manuscript is now with our production department. 

Kind regards, 

on behalf of

Dr. Jeffrey A. Eble 

Academic Editor

PLOS ONE